# Response to Systemic Therapies in Ovarian Adult Granulosa Cell Tumors: A Literature Review

**DOI:** 10.3390/cancers14122998

**Published:** 2022-06-17

**Authors:** Geertruid J. Brink, Jolijn W. Groeneweg, Lotty Hooft, Ronald P. Zweemer, Petronella O. Witteveen

**Affiliations:** 1Department of Gynaecological Oncology, UMC Utrecht Cancer Center, University Medical Center Utrecht, Utrecht University, 3584 CX Utrecht, The Netherlands; g.j.brink-7@umcutrecht.nl (G.J.B.); j.w.groeneweg-11@umcutrecht.nl (J.W.G.); r.zweemer@umcutrecht.nl (R.P.Z.); 2Cochrane Netherlands, Julius Center for Health Sciences and Primary Care, University Medical Center Utrecht, Utrecht University, 3584 CX Utrecht, The Netherlands; l.hooft@umcutrecht.nl; 3Department of Medical Oncology, UMC Utrecht Cancer Center, University Medical Center Utrecht, Utrecht University, 3584 CX Utrecht, The Netherlands

**Keywords:** ovarian cancer, granulosa cell tumor, systemic therapy, chemotherapy, anti-hormonal therapy, systematic review

## Abstract

**Simple Summary:**

Adult granulosa cell tumors (aGCTs) are a rare subtype of ovarian cancer. First choice of treatment is surgery; when this is not possible, chemotherapy and anti-hormonal therapy are often used. There is limited evidence on the effect of systemic therapy in aGCT. The aim of our systematic review is to provide an overview of the response to chemotherapy and anti-hormonal therapy in patients with aGCT. We found very few articles reporting the response to chemotherapy and anti-hormonal therapy in only aGCT. The available data showed a moderate response to chemotherapy and anti-hormonal therapy, but if patients who achieve stable disease are also taken into account, the response is higher. This may mean that surgery can be postponed for a longer period of time.

**Abstract:**

For adult granulosa cell tumors (aGCTs), the preferred treatment modality is surgery. Chemotherapy and anti-hormonal therapy are also frequently used in patients with recurrent aGCT. We aimed to review the existing literature on the response to chemotherapy and anti-hormonal therapy in patients with aGCT. Embase and MEDLINE were searched from inception to November 2021 for eligible studies. Objective response rate (ORR) was calculated as the total number of cases with a complete response (CR) or a partial response (PR). Disease control rate (DCR) was defined as the sum of cases with CR, PR or stable disease (SD). A total of 10 studies were included that reported on chemotherapy and 13 studies were included that reported on anti-hormonal therapy. The response rates of the 56 chemotherapy regimens that could be evaluated resulted in an ORR of 30% and DCR of 58%. For anti-hormonal therapy, the results of 73 regimens led to an ORR of 11% and a DCR of 66%. Evidence on systemic therapy in aGCT only is limited. For both chemotherapy and anti-hormonal therapy, the ORR is limited, but the response is considerably higher when patients achieving SD are included. New approaches are needed to provide more evidence and standardize treatment in aGCT.

## 1. Introduction

Granulosa cell tumors (GCT) of the ovary belong to the rare subgroup of sex cord-stromal cell tumors (SCST), representing 2–3% of all ovarian cancers [1]. The vast majority is of the adult type (aGCT), while the juvenile subtype accounts for 5% of GCTs. In contrast to the juvenile type, the adult type usually presents in the sixth decade and harbors a *FOXL2* somatic mutation in 97% of cases [1,2,3]. Typically, aGCT patients present with complaints of postmenopausal bleeding or abdominal pain, with ultrasound examination showing an adnexal mass as well as endometrial thickening in a subset of patients. In most cases however, the diagnosis of aGCT is only made postoperatively following surgical removal of an ovarian tumor. Although 5-year survival rates are generally favorable, recurrent disease is seen in approximately 50% of aGCT patients and survival rates decline to 66.8% at 20 years [4]. Surgery is the preferred treatment for primary as well as recurrent aGCT. Patients with recurrent disease often develop multiple relapses, requiring repeated debulking surgery with significant morbidity. When surgery is not a viable option, alternative treatment options include chemotherapy and anti-hormonal therapy. Current guidelines recommend a treatment with bleomycin, etoposide and cisplatin (BEP) or carboplatin and paclitaxel (CP) as the gold standard for inoperable recurrent SCST [5,6,7].

These guidelines however, are supported by limited evidence and are primarily based on studies in SCST [8,9,10,11,12] or studies performed in both juvenile and adult type GCT [13,14]. Treatment regimens for juvenile GCT differ because preserving fertility plays an important role in its treatment. Treatment responses in other SCSTs, for example Sertoli–Leydig tumors, cannot be compared one on one to aGCT, as for these tumors, systemic therapy is rarely necessary in a non-adjuvant setting [15].

Previously, Van Meurs, et al. [16] reviewed the existing literature on chemotherapy in GCT, including granulosa theca cell tumors and possibly juvenile GCT or unclassified SCST as well. Their review showed a response rate of 50% (95% confidence interval, 44–57%) for chemotherapy. They concluded that available data were limited and the quality of the reviewed studies was poor due to a lack of standardized response evaluation. 

The use of chemotherapy can be limited by side effects of the recommended regimens. Treatment with BEP is known for its bleomycin-induced pulmonary injury and its toxicity, especially in elderly patients [17]. Treatment with CP can lead to hair loss, sensory neuropathy and dose-dependent bone marrow suppression [18]. As in other cancers, side effects must be weighed against the response to palliative chemotherapy.

Anti-hormonal therapy represents an alternative therapeutic strategy for recurrent aGCT. Granulosa cells produce hormones such as estradiol, inhibin and anti-Müllerian hormone, which are used as tumor markers for aGCT [1,19]. Anti-hormonal therapy is frequently used when the tumor is positive for the estrogen receptor (ER) or progesterone receptor (PR). Two previous studies found that ER and PR were positive in 32–66% and 98–100% of aGCTs, respectively [20,21]. Anti-hormonal therapy is generally better tolerated than chemotherapy and can be administered over a prolonged period. The response to anti-hormonal therapy in aGCT has been evaluated in a previous systematic review that also included juvenile GCT, gynandroblastoma and granulosa theca cell tumors [22]. The authors suggested that anti-hormonal therapy could be a first-line treatment for a subset of patients, based on tumor hormone receptor status. Overall, they concluded that study quality is poor and limited data are available [22]. 

The preference for either chemotherapy or anti-hormonal therapy varies widely among practitioners. This large variation in systemic therapies is maintained by the current guidelines, lacking evidence for the use of systemic therapy in aGCT alone. In order to reliably assess the response to chemotherapy and anti-hormonal therapy in aGCT, available data should not be mixed with results in other SCST subtypes. With this systematic review, we aim to provide an overview of the existing literature on the response rates of both chemotherapy and anti-hormonal therapy in aGCT only. 

## 2. Materials and Methods

This systematic review is registered in the International Prospective Register of Systematic Reviews (PROSPERO ID CRD42022241611) and was performed in accordance with the Preferred Reporting Items for Systematic Reviews and Meta-Analysis (PRISMA) guidelines [23,24]. 

Existing literature on systemic treatment in aGCT was searched in Embase and MEDLINE through PubMed from inception to 9 November 2021. We used the following search terms: “sex cord-stromal tumor”, “granulosa cell tumor”, “chemotherapy” or “anti-hormonal therapy”, with synonyms, alternative spellings and abbreviations of these terms. The detailed search strategy can be found in Appendix A. References of relevant articles were screened for additional studies. 

### 2.1. Study Selection

Titles and abstracts were screened for relevance by two reviewers (J.W.G. and G.J.B.) independently. Potentially relevant studies were retrieved for full-text review. Studies were screened using Rayyan, a web application for filtering eligible articles [25]. Studies were included if they met the preset criteria: English language, histology of adult GCT and evaluation of treatment response based on CT or MRI imaging. Response criteria had to be described or dimensions of the tumor locations had to be noted exactly over time to be included. When studies investigated systemic therapy in several SCST or GCT subtypes but response for aGCT cases was separately reported, those cases were included. Studies on treatment in an adjuvant setting and narrative reviews were excluded. Disagreements between reviewers were resolved by discussion and consensus.

### 2.2. Data Extraction

The following variables from each included study on the use of chemotherapy were extracted: first author, year of publication, study period, study design, number of patients with aGCT, age at inclusion, previous treatments, FIGO stage at diagnosis, number of chemotherapy regimens, dose and type of chemotherapy, side effects, dimensions of tumor locations or reported response defined by World Health Organization (WHO) criteria or Response Evaluation Criteria in Solid Tumors (RECIST) criteria, PFS, overall survival (OS), follow-up time (FU) and disease status. 

The following data were extracted from the studies reporting on anti-hormonal therapy: first author, year of publication, study period, study design, number of patients with aGCT, age at inclusion, previous treatments, FIGO stage at diagnosis, dose and type of anti-hormonal therapy, side effects, dimensions of tumor locations or reported response defined by World Health Organization (WHO) criteria or Response Evaluation Criteria in Solid Tumors (RECIST) criteria, PFS, OS, FU and disease status.

Study period was defined as the period wherein the study cohort was collected; for case reports, the study period was the period during which chemotherapy or anti-hormonal therapy was administered. Previous treatments were defined as any treatment for primary or recurrent aGCT prior to the study treatment. 

Our primary outcome is the response, defined following the RECIST criteria; if response was not specified, the authors of this article calculated the response themselves using the dimensions of the tumor locations [26]. Secondary outcomes are the objective response rate (ORR) and the disease control rate (DCR). The ORR was calculated, using the response as reported, as the sum of cases with complete response (CR) or partial response (PR) and was defined as the percentage of patients whose tumor burden decreases over a certain time period. The DCR was calculated as the sum of CR, PR or stable disease (SD) cases and was defined as the percentage of patients whose tumor burden decreases or remains stable over a certain time period. Both ORR and DCR were calculated to also assess the response rates when including patients with SD, as achieving SD may be relevant in the treatment of aGCT. Disease status was reported in four categories: no evidence of disease (NED), alive with disease (AWD), death of disease (DOD) or death of other cause (DOC).

The extracted data were verified by a second author (J.W.G.) and disagreements were resolved by consensus. When further clarification or additional data were required, the investigators of the study in question were contacted by email. 

### 2.3. Data Analysis

Quality assessment of all included studies was performed using the appropriate tool for each study design and was appraised by two reviewers (J.W.G. and G.J.B.) independently. The RoB 2 tool was used to assess the risk of bias in randomized trials [27], the Newcastle–Ottawa Scale (NOS) [28] was used for case-controlled and cohort studies, and Murad’s tool [29] was used for case series and case reports. Descriptive statistics were used to summarize characteristics and results of the included studies. Treatment response was the main outcome of interest, as described in the article or calculated by the authors following RECIST criteria. Other outcomes of interest included PFS, OS, FU and disease status.

## 3. Results

### 3.1. Study Selection

The search yielded 3081 potentially eligible studies. By assessing article references, three additional studies were found. After removing duplicates, 2817 titles and abstracts were screened for relevance. After reviewing the full text of 100 remaining articles, 22 studies met the eligibility criteria as shown in the PRISMA 2020 Flow diagram (Figure 1).

Out of the 22 included studies, 9 studies described the efficacy of chemotherapy and 12 studies reported the use of anti-hormonal therapy in aGCT. One article studied the response to chemotherapy as well as anti-hormonal therapy in aGCT [30].

### 3.2. Study Characteristics

Of the included studies reporting on chemotherapy, there were two case reports, seven cohort studies and one randomized controlled trial (RCT). A subset of these studies described the response to chemotherapy in all types of SCST or both adult and juvenile GCT, and the results in specifically aGCT could not be retrieved [11,13,31,32]. These studies were therefore excluded from further analysis. An overview of the included studies on chemotherapy in aGCT can be found in Table 1.

Of the 13 studies on anti-hormonal therapy, 5 were cohort studies and 8 were case reports. The included studies are summarized in Table 2. Banerjee, et al. [33] showed the response to anastrozole in 38 GCT patients; however, no distinction was made between juvenile and adult subtypes. This study was therefore excluded from further analysis.

**Table 1 cancers-14-02998-t001:** Summary of studies describing the use of chemotherapy in aGCT.

Author, Year	Study Design	Study Period	Patients (*n*)	Stage at Diagnosis	Previous Treatment *n* (%)	Chemotherapy *n* (%)	Response *n* (%)	PFS (mo) Median (Range)	OS (mo) Median (Range)	FU (mo) Median (Range)	Disease Status *n* (%)
Tresukosol, 1995 [34]	retrospective case report	1992–1994	1	IC ^1^	S ×1, CT ×1	paclitaxel 1(100)	PR 1 (100)	12	24+ ^1^	24 ^1^	AWD 1 (100) ^1^
Shavit, 2012 [35]	retrospective case report	2006–2007	1	IA	S ×2, CT ×2	docetaxel 1 (100)	SD 1 (100)	24	NA	24	NA
Uygun, 2003 [36]	retrospective cohort	1979–1999	4	IIIB-IV	S 4 (100) CT 4 (100)	CC 3 (75) CAP 1 (25)	CR 2 (50) PR 2 (50)	38 (21–73) ^1^	40.5 (33–73) ^1^	40.5 (33–73) ^1^	NED 2 (50) DOD 1 (25) DOC 1 (25)
Pectasides, 2008 [37]	retrospective cohort	1983–2007	5	IA-IV	S 5 (100) CT 5 (100)	CP 2 (40) CVB 1 (10) 5FU 2 (40)	CR 2 (40) PR 1 (20) PD 2 (40)	7 (0–31) ^1^	28 (4–31) ^1^	NA	AWD 4 (80) DOD 1 (20)
van Meurs, 2014 [16]	retrospective cohort	1968–2011	9	I-IIIC	S 9 (100) RT 1 (11) AHT 1 (11)	BEP 9 (100)	CR 1 (11) PR 1 (11) SD 7 (78)	12 (2–50)	50 (4–165)	NA	NED 2 (22) AWD 3 (33) DOD 3 (33) DOC 1 (11)
Wilson, 2015 [30]	retrospective cohort	1955–2012	17 ^2^	IA-IC	S 17 (100) CT ns, RT ns	CT 17 (100)	CR 1 (3) PR 8 (27) SD 4 (13) PD 15 (50) ×1 ^3^	8.6	NA	NA	NA
Brown, 2004 [11]	retrospective cohort	1985–2002	21aGCT /30SCST	IA-IIIC	S 30 (100) CT 22 (73) RT 2 (7)	NPT 17 (57) PT 13 (43)	CR 3 (10) PR 7 (23) SD 7 (23) PD 12 (40) ^4^	16.8 (0–68)	NA	100.7 (8.1–361.3)	NED 3 (10) AWD 20 (67) DOD 5 (17) DOC 2 (6)
Pautier, 2008 [13]	prospective cohort	1990–2002	14aGCT /20GCT	I-IV	S 20 (100) CT 1 (5)	BEP 20 (100)	CR 9 (45) PR 9 (45) SD 1 (5) PD 1 (5)	24 (4–84)	46	45 (3–112)	NED 9 (45) AWD 3 (15) DOD 8 (40)
Burton, 2016 [31]	prospective cohort	2000–2013	31 SCST	NA	S 31 (100) CT 24 (77) RT 3 (10) AHT 3 (10) IT 1 (3)	paclitaxel 31 (100)	CR 1(3) PR 8 (26) SD 15 (48) PD 6 (19) ^5^	10	73.6	67	AWD 15 (48) DOD 16 (52)
Ray-Coquard, 2020 [32]	prospective RCT	2013–2020	27aGCT /32 SCST	I-IV	S 32 (100) CT 32 (100) RT 4 (13) AHT 8 (25)	paclitaxel 32 (100)	CR 0 (0) PR 8 (25) SD 17 (53) PD 7 (22)	14.7 (95% CI 11.5–18.3)	NA	38.9 (IQR 36.4–43.8)	AWD 26 (81) DOD 6 (19)

^1^ Calculated by authors of this paper; ^2^ 30 regimens; ^3^ two patients did not have response recorded; ^4^ one patient who died as a result of unrelated medical causes whose response could not be assessed; ^5^ one patient (3%) was indeterminate; PFS: progression-free survival, OS: overall survival, FU: follow-up in time, RCT: randomized controlled trial, NA: not available, ns: not specified; response according to RECIST criteria: CR: complete response, PR: partial response, SD: stable disease, PD: progressive disease; disease status: NED: no evidence of disease, AWD: alive with disease, DOD: death of disease, DOC: death of other cause; treatments: 5FU: 5-fluorouracil, AHT: anti-hormonal therapy, BEP: bleomycin-etoposide-cisplatin, CAP: cyclophosphamide-doxorubicin-cisplatin, CC: cyclophosphamide-cisplatin; CP: carboplatin-paclitaxel, CT: chemotherapy, CVB: cisplatin-vinblastine-bleomycin, IT: immunotherapy, NPT: non–platinum containing taxane regimens, PT: platinum-containing taxane regimens, RT: radiotherapy, S: surgery.

**Table 2 cancers-14-02998-t002:** Summary of studies describing the use of anti-hormonal therapy in aGCT.

Author, Year	Study Design	Study Period	Patients (*n)*	Stage at Diagnosis	Previous Treatment *n* (%)	Anti-Hormonal Therapy *n* (%)	Response *n* (%)	PFS (mo) or Median (Range)	OS (mo) Median (Range)	FU (mo) Median (Range)	Disease Status *n* (%)
Fishman, 1996 [38]	retrospective cohort	1991–1996	4	NA	CT 4 (100) AHT 1 (25)	leuprolide acetate 4 (100)	PR 2 (50) SD 2 (50)	8 (3–13+) ^1^	NA	11	AWD 3 (75) DOD 1 (25)
Assi, 2017 [39]	retrospective case report	2013–2016	1	I	S ×1, CT ×2	letrozole 1 (100)	PR 1 (100)	35 ^1^	44+ ^1^	44 ^1^	AWD 1 (100)
Hardy, 2005 [20]	retrospective case report	1999–2004	1	>II	S ×3, CT ×2	megestrol/tamoxifen 1 (100)	CR 1 (100)	60+ ^1^	60+ ^1^	60 ^1^	NED 1 (100)
Abdul Munem, 2012 [40]	retrospective case report	2009–2010	1	NA	S ×3, CT ×3, RT ×1	anastrozole 1 (100)	SD 1 (100)	20+	20+	20	AWD 1 (100) ^1^
AlHilli, 2012 [41]	retrospective case report	2010	1	IA	S ×7, CT ×1, RT ×4	letrozole 1 (100)	PR ^1^	6	NA	6	AWD 1 (100) ^1^
van Meurs, 2015 [42]	retrospective cohort	1979–2013	16 ^2^	I-III	S 16 (100) CT 8 (50) RT 7 (44) AHT 6 (38)	anastrozole 2 (9) goserelin 2 (9) letrozole 6 (27) megestrol acetate 6 (27) tamoxifen 5 (23) aromatase inhibitor 1 (5)	SD 14 (64) PD 8 (36)	4 (2–53) ^1^	NA	NA	NED 1 (6) AWD 8 (50) DOD 5 (31) DOC 2 (13)
Wilson, 2015 [30]	retrospective cohort	1955–2012	26 ^3^	IA-IC	S 26 (100) CT ns, RT ns	AHT 126 (100)	CR 1 (2) PR 5 (11) SD 21 (48) PD 12 (27) ^4^	18 (6–54)	NA	NA	NA
Lamm, 2016 [43]	retrospective case report	2012	1	IA ^1^	S ×6, AHT ×1	letrozole 1 (100)	CR 1 (100)	8	NA	12 ^1^	AWD 1 (100)
Schwartz, 2016 [44]	retrospective case report	2008–2009	1 ^5^	IA	S ×2, CT ×1, RT ×1	anastrozole 1 (100)	PR	19 (8–30)	NA	37.5 (30–45)	AWD 1 (100)
Yazigi, 2016 [45]	retrospective case report	2003–2014	1	NA	S ×5, CT ×2, AHT ×2	letrozole 1 (100)	PR ^1^	11	31	31	DOD 1 (100)
Tsubamoto, 2019 [46]	retrospective cohort	2007–2015	3	NA	S 3 (100)	leuprolide acetate 3 (100)	SD 3 (100)	4 (4–22)	27 (6–74)	27 (6–74)	AWD 1 (33) DOD 2 (67)
Moon, 2021 [47]	retrospective case report	NA	1	NA	S ×3, CT ×3, AHT ×2, RT ×1	megestrol acetate/tamoxifen 1 (100)	SD^1^	22	NA	48 ^1^	AWD
Banerjee, 2021 [33]	retrospective cohort	2012–2017	41 GCT	NA	S 41 (100) CT 16 (39) RT 5 (13) ns 19 (46)	anastrozole 38 (100)	PR 1 (3) SD 29 (76) PD 8 (21)	8.6 (95% CI 5.5–13.5)	NA	52	NA

^1^ Calculated by authors of this paper; ^2^ 22 regimens; ^3^ 44 regimens; ^4^ For the remaining 5 patients response could not be assessed; ^5^ 2 regimens; PFS: progression free survival, OS: overall survival, FU: follow-up in time, RCT: randomized controlled trial, NA: not available, ns: not specified; response according to RECIST criteria: CR: complete response, PR: partial response, SD: stable disease, PD: progressive disease; disease status: NED: no evidence of disease, AWD: alive with disease, DOD: death of disease, DOC: death of other cause; treatments: AHT: anti-hormonal therapy, CT: chemotherapy, RT: radiotherapy, S: surgery.

### 3.3. Quality Assessment

The results of the risk of bias and quality assessment for all 22 studies, using the “Risk-of-bias VISualization” tool, are visualized in Appendix B
Figure A1, Figure A2 and Figure A3 [48]. The included RCT was rated as low on overall bias. Of the 11 cohort studies, 8 studies were judged to have low risk, and 3 cohort studies had minor concerns. Finally, 10 case reports were assessed, of which the judgement of ascertainment bias carried more weight. Eight case reports scored a low risk of bias, while two case reports showed concerns on ascertainment bias. Overall, risk of bias scores of all studies was low and all studies were included in the current review.

### 3.4. Chemotherapy

The efficacy of chemotherapy in aGCT was reported by six studies [16,30,34,35,36,37]. Results are summarized in Table 3. Of the 58 evaluated chemotherapy regimens, the ORR was 32% and the cumulative DCR was 60%. When excluding case reports, the remaining four studies together resulted in an ORR of 30% and a DCR of 58%, as shown in Table 4. In order to evaluate the response to different types of chemotherapy, the results were divided into groups with platinum- and/or taxane-containing regimens. Platinum-based chemotherapy includes BEP, cyclophosphamide-doxorubicin-cisplatin, cyclophosphamide-cisplatin and cisplatin-vinblastine-bleomycin. Taxane-based chemotherapy includes docetaxel and paclitaxel. The platinum taxane combination group mainly consisted of CP, and the other regimens comprised 5-fluorouracil (5FU), single agent chlorambucil, doxorubicin and cyclophosphamide. Although there were few regimens to evaluate, the response to platinum-based chemotherapies seemed greater than the response to taxane-based chemotherapies, with a DCR of 68% versus 43%. In the combination group, the DCR was 66%, and for the other therapies, the DCR was 17%. The median PFS could be determined for 21 patients receiving chemotherapy and was 16.5 months (range 0–73 months) with a median OS of 30.5 months (range 3–165 months).

### 3.5. Anti-Hormonal Therapy

The response to anti-hormonal therapy in aGCT was reported by 12 studies and 82 regimens were evaluable, as shown in Table 5 [20,30,38,39,40,41,42,43,44,45,46,47]. The ORR was 19% and the cumulative DCR was 70%. To minimize the possibility of publication bias due to the included case reports, the results of all articles other than case reports are shown in Table 6. Similar to the response rate found when including case reports, the ORR of anti-hormonal therapy was 11% and the calculated DCR was 66%. The most frequently used regimens were aromatase inhibitors (9 regimens) and GnRH agonists (9 regimens), with a DCR of 67% and 89%, respectively. The median PFS following anti-hormonal therapy was given for 23 patients and was 4 months (range 2–53 months) and the median OS was 14 months (3–112 months).

## 4. Discussion

With this systematic review, we provide an overview of the current literature on systemic therapies in aGCT. We identified 10 studies describing the response to chemotherapy, and 13 studies reporting the response to anti-hormonal therapy in aGCT only, as shown in Table 1 and Table 2. When reviewing these studies, 56 regimens of chemotherapy and 73 regimens of anti-hormonal therapy were evaluable. The ORR and DCR for chemotherapy were 30% and 58%, and for anti-hormonal therapy, they were 11% and 66%, respectively. The observed duration of PFS following chemotherapy (0–73 months) or anti-hormonal therapy (2–53 months) for aGCT varies widely. An accurate comparison between the response rates of the different systemic therapy regimens could not be made, due to the low numbers of treatments.

Of interest, a large difference between the ORR and DCR for both chemotherapy and anti-hormonal therapy was observed. These results imply that a large proportion of patients responding to chemotherapy or anti-hormonal therapy achieve stable disease. A response rate that includes stable disease is of relevance in aGCT, offering the clinical benefit of postponing further deterioration or prolonging the interval between surgical treatments, thereby reducing overall morbidity. In addition, the described response rates contradict the previously reported pooled ORRs for chemotherapy and anti-hormonal therapy, which were more favorable: 50% (95% CI, 44–57%) and 71% (95% CI, 52–85%), respectively [16,22]. A DCR was not reported by these studies, but the calculated DCR for chemotherapy and anti-hormonal therapy was 72% and 84%. Their higher response rates may be explained by inclusion of cases with a response determined clinically or surgically/pathologically. Of the 224 evaluable patients, only 86 responses were evaluated based on imaging [16]. Another explanation for this inconsistency in ORR could be the inclusion of case reports, potentially leading to publication bias [22]. In addition, both reviews not only included patients with aGCT but also juvenile GCT, gynandroblastoma, granulosa theca cell tumors and possibly other SCST subtypes. The comparison of these results with our findings emphasizes the importance of including the ORR and separating the data of other subtypes from the results in aGCT.

Another important finding of the current review is the limited amount of data available on the use of chemotherapy and anti-hormonal therapy in aGCT specifically. Only 22 studies reported separate response rates for aGCT, including 56 evaluable regimens for chemotherapy and 73 evaluable regimens for anti-hormonal therapy. Studies with the largest patient numbers had to be excluded, because response rates were not reported separately for aGCT. The current guidelines are mainly based on these excluded studies, implying that the majority of the evidence supporting these guidelines is based on data of any SCST subtype [5,6,7]. Other SCST subtypes include juvenile GCT, Sertoli (–Leydig) cell tumors, gynandroblastoma and sex cord tumor with annular tubules, which behave differently and may respond differently to treatments [49,50]. These differences make it important to differentiate the results in aGCT from those in other SCST subtypes, so that the true response in aGCT can be determined. In the future, this may allow us to predict the response rate of a particular treatment for aGCT patients, thus avoiding unnecessary treatment-related morbidity. Based on our findings, response rates of chemotherapy in aGCT may previously have been overestimated. Our results furthermore show that many patients achieve stable disease with anti-hormonal therapy, which has far fewer side effects. Thus, it can be suggested that chemotherapy should be administered more cautiously in aGCT, and anti-hormonal therapy could be considered as the first choice systemic treatment in a subset of patients with recurrent aGCT.

Strengths of this study include its thorough search strategy and the inclusion of individual aGCT cases from cohort studies. This way, we aimed to maximize the number of evaluable aGCT cases. Limitations of our review include the use of definitions such as PFS and OS that were defined differently among the reviewed studies, making them more difficult to interpret. Moreover, case reports were included in this review. To reduce the risk of bias, the response rates of the different systemic treatments were also shown without the data collected from case reports. Finally, the time to the next treatment for recurrence was rarely described in the reviewed studies and could therefore not be reported. The duration of this period is relevant, because patients with recurrent aGCT often can be managed initially by “watchful waiting” and in the absence of symptoms, treatment may only become necessary at a later moment. Therefore, knowing not just the duration of the PFS but also the time until the next treatment would be clinically meaningful.

In order to truly assess the response to chemotherapy and anti-hormonal therapy in aGCT, novel research approaches are needed. A randomized controlled trial in aGCT, like the one recently initiated by the GOG, requires many years to include a sufficient number of patients. Examples of novel approaches include a master protocol [51], such as the BASKET trial performed by How et al. [52], and in vitro drug screens. Haltia et al. [53] previously tested many different therapies using drug screens, and Roze et al. [54] recently demonstrated the use of patient-derived aGCT cell lines for drug screens. Further research could comprise a study with the design of an umbrella trial [51]. Such a trial offers the opportunity to test different therapies in one disease type. A possible design of an umbrella trial for aGCT could consist of patients with recurrent aGCT who are initially evaluated for operability. If they are considered inoperable, anti-hormonal therapy or chemotherapy could be initiated depending on the trial arm. This way, the response to different systemic therapies can be compared, thereby standardizing the therapeutic options for recurrent aGCT.

## 5. Conclusions

To conclude, this review demonstrates that the existing literature on response rates of chemotherapy and anti-hormonal therapy in specifically aGCT is scarce. Current guidelines advise practitioners to treat inoperable, recurrent aGCT with chemotherapy, suggesting both BEP and CP regimens. Notwithstanding the relatively limited data, for both chemotherapy and anti-hormonal therapy the DCR is considerably higher than the ORR, implying that systemic treatment leads to SD in a substantial proportion of patients. The importance of anti-hormonal therapy may be especially relevant, considering that 55% of the patients achieved stable disease with anti-hormonal therapy, which may be clinically valuable. In patients with low tumor load and few complaints, anti-hormonal therapy could be the first choice. Novel research approaches need to be designed in order to strengthen the current evidence and further develop and standardize treatment options for aGCT.

## Figures and Tables

**Figure 1 cancers-14-02998-f001:**
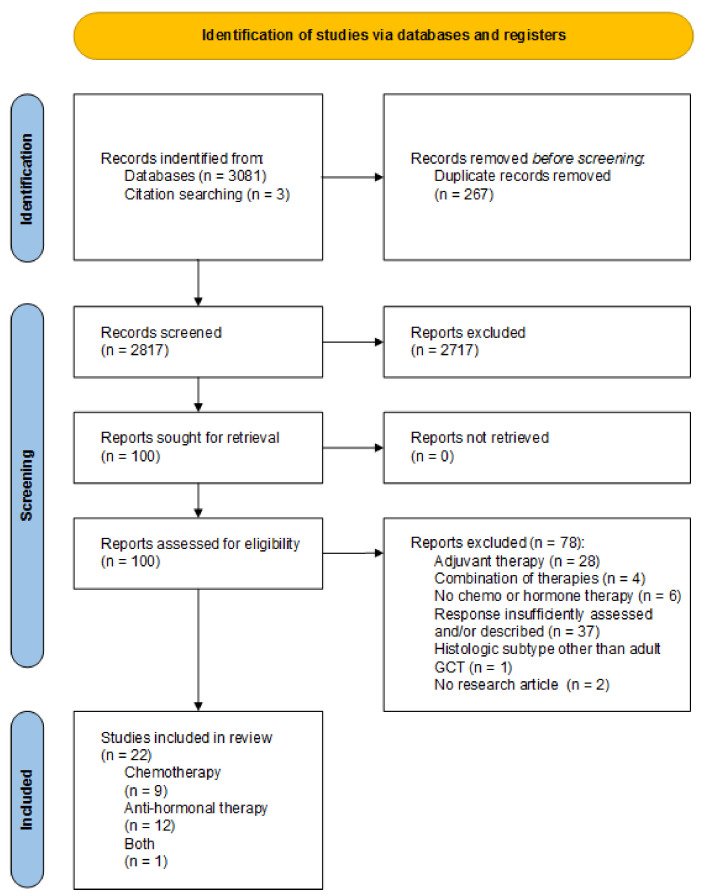
PRISMA Flow diagram, summarizing search results.

**Table 3 cancers-14-02998-t003:** Summary of response rates of chemotherapy for aGCT.

Regimen	Number of Regimens	CR *n* (%)	PR *n* (%)	SD *n* (%)	PD *n* (%)	Unknown *n* (%)
Platinum-based	37	4 (11)	11 (30)	10 (27)	10 (27)	2 (5) ^1^
Taxane-based	9	0 (0)	2 (22)	3 (33)	4 (45)	0 (0)
Platinum taxane combination	6	2 (33)	0 (0)	2 (33)	2 (33)	0 (0)
Other ^2^	6	0 (0)	0 (0)	1 (17)	5 (83)	0 (0)
**Total**	**58**	**6 (10)**	**13 (22)**	**16 (28)**	**21 (36)**	**2 (4)**

^1^ One response not stated, one patient stopped due to toxicity; ^2^ including 5-fluorouracil, chlorambucil, doxorubicin and cyclophosphamide. CR: complete response, PR: partial response, SD: stable disease, PD: progressive disease.

**Table 4 cancers-14-02998-t004:** Summary of response rates of chemotherapy for aGCT, excluding case reports.

Regimen	Number of Regimens	CR *n* (%)	PR *n* (%)	SD *n* (%)	PD *n* (%)	Unknown *n* (%)
Platinum-based	37	4 (11)	11 (30)	10 (27)	10 (27)	2 (5) ^1^
Taxane-based	7	0 (0)	1 (14)	2 (29)	4 (57)	0 (0)
Platinum taxane combination	6	2 (33)	0 (0)	2 (33)	2 (33)	0 (0)
Other ^2^	6	0 (0)	0 (0)	1 (17)	5 (83)	0 (0)
**Total**	**56**	**5 (9)**	**12 (21)**	**16 (28)**	**22 (39)**	**2 (3)**

^1^ One response not stated, one patient stopped due to toxicity; ^2^ including 5-fluorouracil, chlorambucil, doxorubicin and cyclophosphamide. CR: complete response, PR: partial response, SD: stable disease, PD: progressive disease.

**Table 5 cancers-14-02998-t005:** Summary of response rates of anti-hormonal therapy for aGCT.

Regimen	Number of Regimens	CR *n* (%)	PR *n* (%)	SD *n* (%)	PD *n* (%)	Unknown *n* (%)
Aromatase inhibitor	16	1 (6)	5 (31)	7 (44)	3 (19)	0 (0)
GnRH agonist	9	0 (0)	2 (22)	6 (67)	1 (11)	0 (0)
Progestin	6	0 (0)	0 (0)	5 (83)	1 (17)	0 (0)
SERM	5	0 (0)	0 (0)	2 (40)	3 (60)	0 (0)
Combinations	2	1 (50)	0 (0)	1 (50)	0 (0)	0 (0)
Type unknown	44	1 (3)	5 (11)	21 (48)	12 (27)	5 (11) ^1^
**Total**	**82**	**3 (4)**	**12 (15)**	**42 (51)**	**20 (24)**	**5 (6)**

^1^ For the remaining five patients, one was lost to follow-up, two were within the first 6 months of treatment and two did not tolerate treatment. CR: complete response, PR: partial response, SD: stable disease, PD: progressive disease, GnRH: gonadotropin-releasing hormone, SERM: selective estrogen receptor modulator.

**Table 6 cancers-14-02998-t006:** Summary of response rates of anti-hormonal therapy for aGCT, excluding case reports.

Regimen	Number of Regimens	CR *n* (%)	PR *n* (%)	SD *n* (%)	PD *n* (%)	Unknown *n* (%)
Aromatase inhibitor	9	0 (0)	0 (0)	6 (67)	3 (33)	0 (0)
GnRH agonist	9	0 (0)	2 (22)	6 (67)	1 (11)	0 (0)
Progestin	6	0 (0)	0 (0)	5 (83)	1 (17)	0 (0)
SERM	5	0 (0)	0 (0)	2 (40)	3 (60)	0 (0)
Type unknown	44	1 (3)	5 (11)	21 (48)	12 (27)	5 (11) ^1^
**Total**	**73**	**1 (1)**	**7 (10)**	**40 (55)**	**20 (27)**	**5 (7)**

^1^ For the remaining five patients, one was lost to follow-up, two were within the first 6 months of treatment and two did not tolerate treatment. CR: complete response, PR: partial response, SD: stable disease, PD: progressive disease, GnRH: gonadotropin-releasing hormone, SERM: selective estrogen receptor modulator.

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
