# Peer review of "Response to Systemic Therapies in Ovarian Adult Granulosa Cell Tumors: A Literature Review"

_cancers, 2022, doi:10.3390/cancers14122998_

Round 1

Reviewer 1 Report

This article provided an overview of the systematic therapies in adult granulosa cell tumors (aGCTs). The major treatment of aGCTs has been surgery, and the authors collected the information from recent literature and summarized this systematic review. The literature is mostly very recent, and the design of these studies and the data analysis are reasonable. Tables are easy to interpret the efficiency of chemotherapy and anti-hormonal therapy. Although the samples are still limited, the summarized data indicated that anti-hormonal therapy could be the first choice. Their results give valuable insights into their community and also will benefit the clinical practice in the future. I recommend this manuscript to publish in Cancers with the following minor revision.

·       Line 69: “Previously, Van Meurs, et al. [16] reviewed the existing literature on chemotherapy in GCT, including granulosa theca cell tumors and possibly juvenile GCT or unclassified SCST as well.[16]……..standardized response evaluation. [16]”  You don’t need to refer the same reference [16] three times.

·       In 3.4 Chemotherapy, Line 236: “The efficacy of chemotherapy in aGCT was reported by six studies.” It could be better to add reference here so that the reader could check the literature easily. 

·       In 4. Discussion, Line 295: “We identified 10 studies describing the response to chemotherapy and 13 studies reporting the response to anti-hormonal therapy in aGCT only.”  Are they shown in Table 1 and 2? If so, please indicate each Table in the text body.

·       Line 310: 71% (95% CI, 52-85)  85% (% missing)

Reviewer 2 Report

Dear authors,

This systematic review focused to response to chemotherapy and anti-hormonal therapy in adult patients with granulosa cell tumors. The introduction provides sufficient background and includes relevant references, the cited references are relevant to the research, the research design is appropriate, the methods are adequately described, the results are clearly presented, and the conclusions are supported by the results. Only minor changes are necessary before publication. First, the authors should review the placement of punctuation marks throughout the manuscript, and second, the discussion could be more in-depth, and include references that exist and are relevant to the topic of study.
